# SEINE: SHORT-TO-LONG VIDEO DIFFUSION MODEL FOR GENERATIVE TRANSITION AND PREDICTION

**Xinyuan Chen**[1][*] **Yaohui Wang**[1][*] **Lingjun Zhang**[2,1][‡] **Shaobin Zhuang**[1,3]
**Xin Ma**[4,1][‡] **Jiashuo Yu**[1] **Yali Wang**[5,1] **Dahua Lin**[1][†] **Yu Qiao**[1][†] **Ziwei Liu**[6][†]

[1] Shanghai Artificial Intelligence Laboratory, [2] East China Normal University
[3] Shanghai Jiao Tong University, [4] Dept of Data Science & AI, Monash University
[5] Shenzhen Institute of Advanced Technology, Chinese Academy of Sciences
[6] S-Lab, Nanyang Technological University

## ABSTRACT

Recently video generation has achieved substantial progress with realistic results. Nevertheless, existing AI-generated videos are usually very short clips ("shot-level") depicting a single scene. To deliver a coherent long video ("story-level"), it is desirable to have creative transition and prediction effects across different clips. This paper presents a short-to-long (S2L) video diffusion model, **SEINE**, that focuses on generative transition and prediction. The goal is to generate high-quality long videos with smooth and creative transitions between scenes and varying lengths of shot-level videos. Specifically, we propose a random-mask video diffusion model to automatically generate transitions based on textual descriptions. By providing the images of different scenes as inputs, combined with text-based control, our model generates transition videos that ensure coherence and visual quality. Furthermore, the model can be readily extended to various tasks such as image-to-video animation and auto-regressive video prediction. To conduct a comprehensive evaluation of this new generative task, we propose three assessing criteria for smooth and creative transition: temporal consistency, semantic similarity, and video-text semantic alignment. Extensive experiments validate the effectiveness of our approach over existing methods for generative transition and prediction, enabling the creation of story-level long videos. Project page: https://vchitect.github.io/SEINE-project/.

**Animation:** *a dog in spacesuit.*

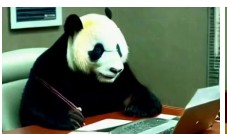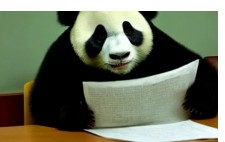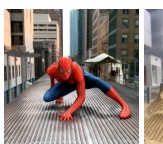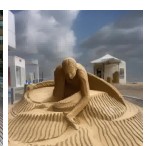

**Transition:** *The panda is working in the office and reading a paper*

**Transition:** *Spider-man becomes a sand sculpture.*

**Prediction:** *Iron Man flying in the sky.*

Figure 1: **Generated samples.** Our S2L model is able to synthesize high-definition transition (shown in the left two columns) and prediction videos (shown in the right column) by giving textual descriptions. *Best view with Acrobat Reader. Click the images to play the videos.*

[*]Equal contribution. [†] Correspondence. [‡] Work done as interns at Shanghai AI Laboratory.

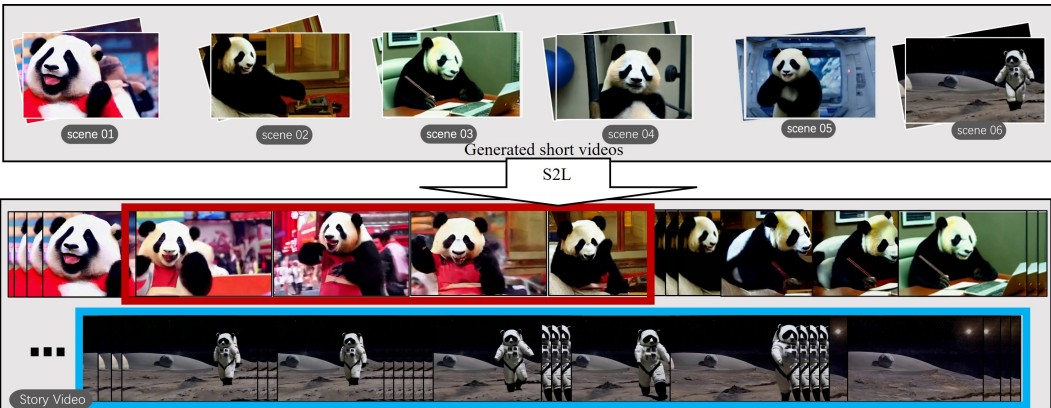

Figure 2: **Short-to-long (S2L) pipeline.** Our S2L video generation model enables the seamless connection of shots from different scenes with transitions and generates videos of varying lengths through autoregressive prediction, facilitating the creation of story-level videos. The red box represents the transition video, while the blue box represents a long video generated through prediction.

## 1 INTRODUCTION

Currently, with the success of diffusion model-based (Ho et al., 2020; Song et al., 2021a;b) text-to-image generation models (Ramesh et al., 2021; 2022; Saharia et al., 2022; Balaji et al., 2022; Rombach et al., 2022), a series of video generation works (Singer et al., 2023; Ho et al., 2022; Blattmann et al., 2023; Zhou et al., 2022; He et al., 2022; Wang et al., 2023b) have emerged and demonstrated impressive results. However, current video generation methods typically only yield "shot-level" video generation, which only consists of around a few seconds and depicts a single scene. Such shot-level short videos fall short of the demands for cinematic and film productions.

In cinematic or industrial-level video productions, "story-level" long video is typically characterized by the creation of distinct shots with different scenes. These individual shots of various lengths are interconnected through techniques like transitions and editing, providing a way for longer video and more intricate visual storytelling. The process of combining scenes or shots in film and video editing, known as transition, plays a crucial role in post-production. Traditional transition methods, including dissolves, fades, and wipes, rely on predefined algorithms or established interfaces. However, these methods lack flexibility and are often limited in their capabilities. An alternative approach to achieve seamless and smooth transitions involves the use of diverse and imaginative shots to smoothly connect between scenes. This technique, commonly employed in films, cannot be directly generated using predefined programs. In this work, we present our model to address the less common problem of generating seamless and smooth transitions by focusing on generating intermediate frames between two different scenes. We establish three criteria that the generated transition frames should meet: 1) semantic relevance to the given scene image; 2) coherence and smoothness within frames; and 3) consistency with the provided text.

In this work, we present a short-to-long (S2L) video diffusion model, **SEINE**, for generative transition and prediction. The goal is to generate high-quality long videos with smooth and creative transitions between scenes and varying lengths of shot-level videos. The pipeline is shown in Fig. 2. We delve into the study of a new task, "generative transition", which aims at producing diverse and creative transition video segments to connect two different scenes. Alongside, we propose an approach to tackle this task and establish criteria for evaluating the efficacy of methods. We approach the problem of transitions as a conditional video generation task. By providing the initial and final frames of a given scene as inputs, we employ a text-based and video conditioning method to generate a smooth transition video in between. This approach allows us to effectively control the transition process and ensure the coherence and visual quality of the generated videos. To this end, we design a flexible random-mask diffusion model. The model is of capable generating transition shots by giving the first frame of one scene and the last frame of another scene, the scene transition is generated by a given textual description. Our model can be extended to image-to-video animation

and autoregressive video prediction, enabling the generation of long-shot videos and dynamic video creation in film production.

We summarize our contributions as follows: **1)** We propose a new problem of generative transition and prediction, aiming at coherent "story-level" long video generation with smooth and creative scene transition and varying lengths of videos; **2)** We present a short-to-long video diffusion model, **SEINE**, focusing on generative transition and prediction; **3)** we propose three assessing criteria for transition and extensive experiments demonstrate our method the superior performance on the metrics, as well as its ability to be applied across versatile applications.

## 2   RELATED WORKS

**Text-to-Video Generation.** In recent years, diffusion models (Ho et al., 2020; Song et al., 2021a;b) have significantly advanced the field of text-to-image (T2I) generation. Existing methods (Ramesh et al., 2021; He et al., 2022; Ramesh et al., 2022; Saharia et al., 2022) have made remarkable progress in generating realistic images that are closely related to textual prompts. Recent advancements in T2I generation have led to the expansion of diffusion models to develop video generation from domain-specific models (Tulyakov et al., 2018; Wang et al., 2020a;b; Tian et al., 2021; Skorokhodov et al., 2022) to general text-to-video (T2V) models. This extension involves adapting the conventional 2D UNet architecture into a spatial-temporal 3D network to establish temporal correlations between video frames. Make-A-Video (Singer et al., 2023) and Imagen Video Ho et al. (2022) leverage text-to-image diffusion models of DALL·E2 (Ramesh et al., 2022) and Imagen (Saharia et al., 2022) respectively for large-scale T2V generation. PYoCo (Ge et al., 2023) presented a noise prior approach and utilized a pre-trained eDiff-I (Balaji et al., 2022) as initialization. An alternative approach involves constructing a T2V model based on pre-trained Stable Diffusion and subsequently fine-tuning the model either entirely (Zhou et al., 2022; He et al., 2022) or partially (Blattmann et al., 2023) on video data. In addition, LaVie (Wang et al., 2023b) fine-tuned the entire T2V model on both image and video datasets. While these methods demonstrate the potential of text-to-video synthesis, they are mostly limited to generating short videos.

**Transition Generation.** Scene transition is a less common problem but is crucial in storytelling, serving as the bridge that connects different narrative moments. They facilitate the smooth progression of a story through shifts in time, place, or perspective. Several techniques are employed to achieve effective scene transitions, such as "fade", "dissolves", "wipes", "IRIS" or simply cut where a scene abruptly transitions to another. These transitions can be achieved using pre-defined algorithms with fixed patterns. Morphing (Wolberg, 1998; Shechtman et al., 2010) enables smooth transitions involving finding pixel-level similarities and estimating their transitional offsets. Existing generative models (Van Den Oord et al., 2017) can leverage linear interpolation at the latent code level to capture semantic similarities. This approach has been widely explored in various works, *e.g.,* style transfer (Chen et al., 2018), object transfiguration (Sauer et al., 2022; Kang et al., 2023).

**Long Video Generation.** Long video generation is a long-standing challenge in the field. Earlier works (Skorokhodov et al., 2022; Yu et al., 2022; Chen et al., 2020) employing generative adversarial networks (GANs) or variational auto-encoder (VAE) to model video distributions. VideoGPT (Yan et al., 2021) employs a combination of VQVAE (Van Den Oord et al., 2017) and transformer architecture to sequentially produce tokens within a discrete latent space. TATS (Ge et al., 2022) trains a time-agnostic VQGAN and subsequently learns a time-sensitive transformer based on the latent features. Phenaki (Villegas et al., 2023) uses a transformer-based model to compress videos into discrete tokens and generate longer continuous videos based on a sequence of text prompts. Recently, Diffusion Models (DMs) (Ho et al., 2020; Song et al., 2021a; Nichol & Dhariwal, 2021) have made significant advancements in video generation. LEO (Wang et al., 2023c) proposed to produce long human videos in latent motion space. LVDM (He et al., 2022) focuses on generating shot-level long videos by employing a hierarchical latent diffusion model. NUWA-XL (Yin et al., 2023) employs a hierarchical diffusion model to generate long videos, but it has only been trained on the cartoon FlintstonesHD dataset due to the lack of datasets on natural videos. On the other hand, Gen-L-Video (Wang et al., 2023a) generates long videos by utilizing existing short text-to-video generation models and treating them as overlapping short video clips. In contrast, our approach views long videos as compositions of various scenes and shot-level videos of different lengths. Our

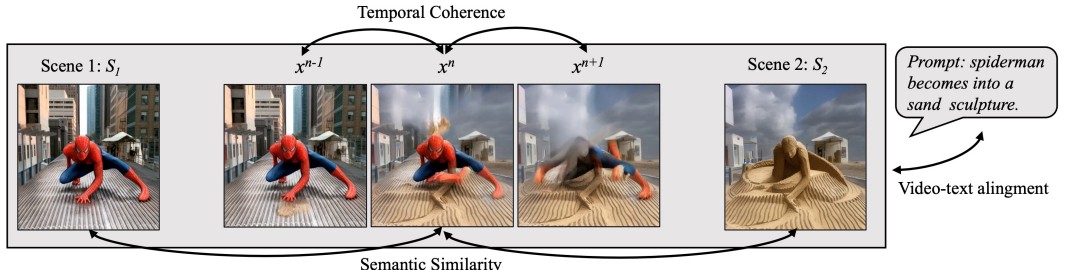

Figure 3: **Notations and schematic representation of generative transition objective.** Generative transition exhibits *semantic similarity* between each frame $x^n$ with the two scene images; each frame $x^n$ and its neighboring frames $x^{n-1}$ and $x^{n+1}$ should be *temporal coherence*; the semantics of generated frames should be *alignment* to the provided textual description.

short-to-long model aims to generate "story-level" videos containing continuous multi-scene videos with varying shot lengths.

## 3 METHODOLOGY

### 3.1 PRELIMINARIES

**Text-to-Image Diffusion Model.** Stable Diffusion (Rombach et al., 2022) performs image diffusion in a compressed latent space of an autoencoder (*i.e.*, VQ-GAN and VQ-VAE). During training, a given image sample $x_0$ is encoded into the latent code $z = \mathcal{E}(x_0)$ and corrupted by Gaussian noise from a pre-defined Markov chain:

$$q(z_t|z_{t-1}) = \mathcal{N}(z_t; \sqrt{1-\beta_t}z_{t-1}, \beta_t I) \tag{1}$$

for $t = 1, \cdots, T$, with $T$ being the number of steps in the forward diffusion process. The sequence of hyperparameters $\beta_t$ determines the noise strength at each step. The above iterative process can be reformulated in a closed-form manner as follows:

$$z_t = \sqrt{\bar{\alpha}_t}z_0 + \sqrt{1-\bar{\alpha}_t}\epsilon, \ \epsilon \sim \mathcal{N}(0, I) \tag{2}$$

where $\bar{\alpha}_t = \prod_{i=1}^t \alpha_t, \alpha_t = 1 - \beta_t$. The model adopts $\epsilon$-prediction and DDPM to learn a function $\epsilon_\theta$ is minimized by:

$$\mathcal{L} = \mathbf{E}_{z_0, c, \epsilon \sim \mathcal{N}(0,I), t}[\|\epsilon - \epsilon_\theta(z_t, t, c)\|_2^2] \tag{3}$$

where $c$ indicates the condition of the textual description, $\epsilon$ is drawn from a diagonal Gaussian distribution. The $\epsilon_\theta$ is commonly implemented by a Unet, where $\theta$ is a parameterized neural network.

**Text-to-Video Diffusion Model.** Our framework is constructed upon a pre-trained diffusion-based T2V model, LaVie (Wang et al., 2023b), which is a cascaded framework that includes a base T2V model, a temporal interpolation model, and a video super-resolution model. LaVie-base, which is part of the LaVie framework, is developed using Stable Diffusion pre-trained model and incorporates a temporal attention module and image-video joint fine-tuning. To benefit from its ability to generate visually and temporally consistent videos with diverse content, we utilize the pre-trained LaVie-base model as the initialization for our framework.

### 3.2 GENERATIVE TRANSITION

The objective of the transition task is to generate a sequence of intermediate frames, denoted as $x^n$, between two given images $S_1$ and $S_2$, using a transition description caption $c$. We define $x^0$ as $S_1$ and $x^N$ as $S_2$, so the sequence consists of $N-2$ target (in-between) frames. The generated sequence should satisfy the following three properties, as shown in Fig. 3: **1) Temporal Coherence**: The changes in visual content across the sequence should be smooth and coherent over time. This property ensures that the generated video appears natural and visually consistent. **2) Semantic Similarity**: The semantic content of each generated intermediate frame should resemble either of

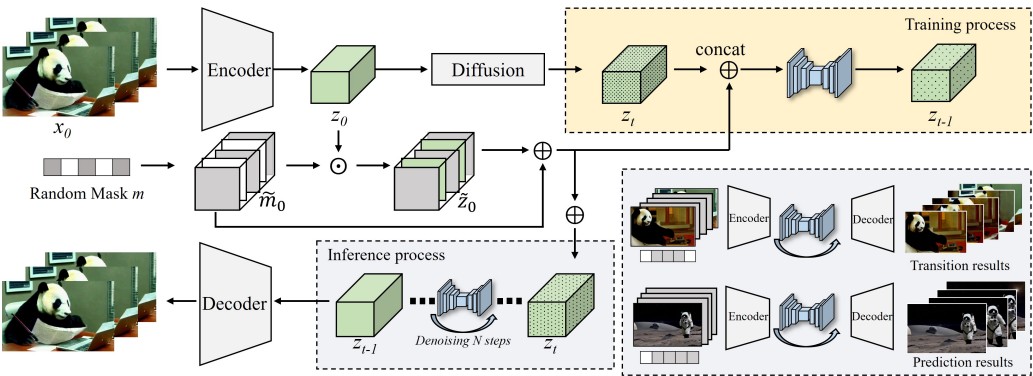

Figure 4: **Overview of our proposed method.** We present our S2L generation model for generating transition video and long video prediction. Our model supports text as input, which is processed through a clip encoder and injected into the model (blue part). For brevity, we omit the details of the text injection, which is identical to the stable diffusion model.

the source images to some extent. This prevents the appearance of the sequence from deviating too much from the source scene images. Additionally, the similarity should gradually change from one source image to the other. **3) Video-text Alignment**: The generated frames should align with the provided text description. This alignment allows us to control the transition between the two source images using the text description.

### 3.3 SHORT-TO-LONG VIDEO DIFFUSION MODEL

We develop a diffusion-based model for S2L videos via transition generation and video prediction. In order to generate unseen frames of transition and prediction, based on visible conditional images or videos, our S2L video diffusion model incorporates a random mask module, as shown in Fig. 4. Given the video dataset $p_{\text{video}}$, we take a $N$-frame original video denoted as $x_0 \in \mathbb{R}^{N \times 3 \times H \times W}$, where $x_0$ follows the distribution $p_{\text{video}}$. The frames of video $x_0$ are first encoded by the pre-trained variational auto-encoder simultaneously as $z_0 \in \mathbb{R}^{n \times c \times h \times w}$, where $c$, $h$ and $w$ indicate the channel, height, and width of the latent code, $n$ represents the frame number. To achieve more controllable transition videos and leverage the capability of short text-to-video generation, our model also takes a textual description $c$ as an input. The goal of our model is to learn a dual condition diffusion model: $p_\theta(z_0|c, \tilde{z}_0)$. In the training stage, the latent code of video is corrupted as Eq. 3: $z_t = \sqrt{\bar{\alpha}_t}z_0 + \sqrt{1 - \bar{\alpha}_t}\epsilon$, where the initial noise $\epsilon \sim \mathcal{N}(0, I)$ whose size is same as $z_0$: $\epsilon \in \mathbb{R}^{n \times c \times h \times w}$. To capture an intermediate representation of the motion between frames, we introduce a random-mask condition layer at the input stage. This layer applies a binary mask $m \in \mathbb{R}^n$ broadcasting to $\tilde{m} \in \mathbb{R}^{n \times c \times h \times w}$ as the size of the latent code, resulting in a masked latent code:

$$\tilde{z}_0 = z_0 \odot m, \tilde{z}_0 \in \mathbb{R}^{n \times c \times h \times w}. \tag{4}$$

The binary masks, represented by $m$, serve as a mechanism to selectively preserve or suppress information from the original latent code. In order to identify which frames are masked and which ones are visible conditional images, our model takes the masked latent code $\tilde{z}_0$ concatenated with the mask $m$ as a conditional input:

$$\tilde{z}_t = [z_t; m; \tilde{z}_0], \tilde{z}_t \in \mathbb{R}^{n \times (3 \times c) \times h \times w}, \tag{5}$$

where $\tilde{z}_t$ represents the final input to the U-Net. The model is trained to predict the noise of the entire corrupted latent code:

$$\mathcal{L} = \mathcal{E}[||\epsilon - \epsilon_\theta(\tilde{z}_t, t, c)||]. \tag{6}$$

This involves learning the underlying distribution of the noise that affects the unmasked frames and textual description. By modeling and predicting the noise, the model aims to generate realistic and visually coherent transition frames that seamlessly blend the visible frames with the unmasked frames. The mask has $n$ elements indicating the index of frames, each of which can take on the values 0 or 1. The probability of the element taking the value 1 is denoted as $p$: m $\sim$ Bernoulli$(p)$. Given that our model primarily focuses on transitions and predictions, where typically less than 3 frames are involved. In our model, we set $p = 0.15$ considering the probability $\frac{2}{n} < p < \frac{3}{n}$. This

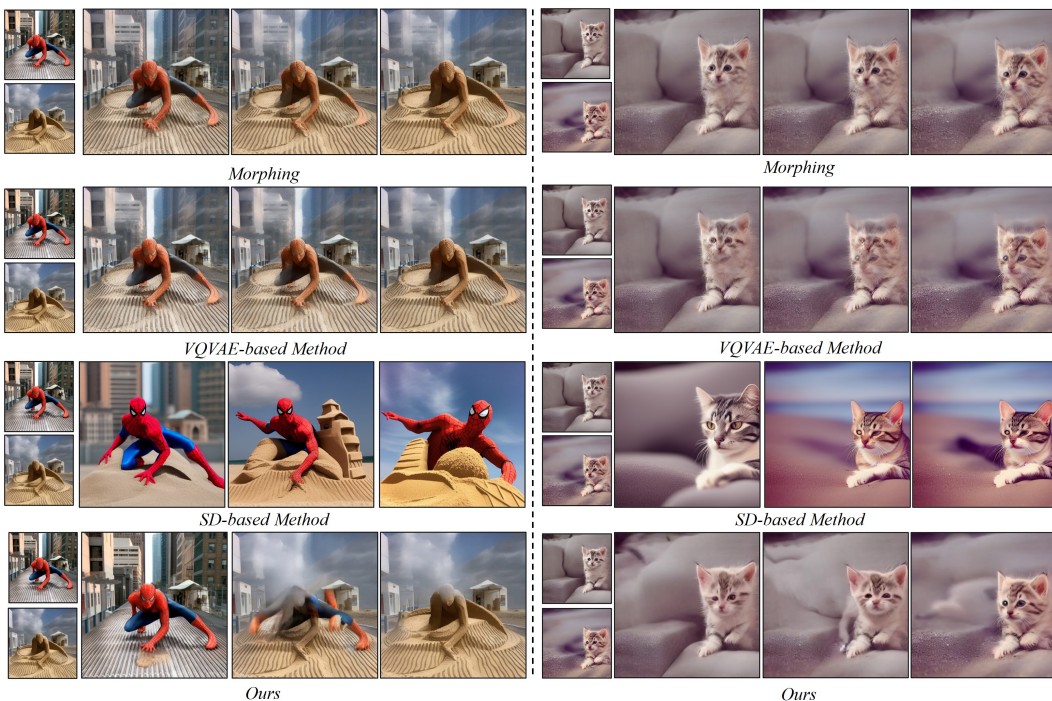

Figure 5: **Qualitative comparisons with existing methods.** Left: *Spiderman becomes into a sand sculpture.* Right: *A cat from sitting on the coach transfer to lying on the sand.*

choice of probability ensures that, on average, approximately two or three out of every $n$ frames will be visible. It aligns with the assumption that the mask is primarily used for capturing transitions between frames, where adjacent frames are more likely to be influenced by the mask. Our random-mask based model is capable of generating frames for any given frames at arbitrary positions within the sequence. **Transition** can be obtained by providing the first and last frames of a sequence and utilizing prompts to control the transition style and content, resulting in intermediate transition frames that depict the transition or progression within the video sequence. **Long video** involves recursive using the last few frames of the generated video as input and utilizing masks to predict the subsequent frames. By recursively iterating this process, the model can generate longer video sequences. **Image-to-video animation** can be achieved by treating the reference image as the first frame. The first binary mask is set as ones and the remaining is set as zeros. The animated video can be obtained through a denoising process and subsequently decoded from the encoder.

## 4 EXPERIMENTS

**Implementation Details.** SEINE is initialized from LaVie-base (Wang et al., 2023b), with the addition of a random mask layer being randomly initialized. We first utilize the WebVid10M dataset (Bain et al., 2021) as the main training set, and during the later stages of training, we use a few internal watermark-free datasets to mitigate the presence of watermarks in the generated output. Our model is trained on videos of $320 \times 512$ resolution with 16 frames. During inference, SEINE is able to generate videos of arbitrary aspect ratios. **Comparison Methods.** Transition video generation is a relatively understudied and novel problem within the field. While traditional methods have explored smooth transitions through image morphing techniques, there is a lack of specific approaches addressing generative transition in the context of video generation. In this part, we conduct analyses between our proposed approach and existing methods that can achieve similar effects: diffusion model-based method (*i.e.*, Stable Diffusion), VQGAN-based model (Esser et al., 2021)). Details description and implementation for the compared methods are described in App. A.1.

## 4.1 QUALITATIVE COMPARISON

Fig. 5 shows the visual comparison with the existing methods. We observe that using interpolation methods for VQVAE to generate intermediate transition scenes often results in a transparent blending of two images. Morphing techniques, on the other hand, can achieve object deformation by matching key points (such as changes in a cat's pose), ensuring that objects do not undergo transparent blending. However, for scenes without significant object deformation, such as transitioning from a cityscape to the sky in a Spiderman scene, simple blending may still occur. Stable Diffusion-based methods do not suffer from simple blending issues, but their lack of sufficient continuity in hidden codes leads to a lack of consistency in the generated intermediate images. In contrast, our method ensures smooth object motion while incorporating plausible phenomena during transitions.

## 4.2 QUANTITATIVE COMPARISON

We assess the generative transition from three aspects: *temporal coherence*, *semantic similarity*, and *video-text alignment*. To evaluate the semantic similarity between generated frames and given scene images, we use the clip similarity (CLIPSIM) metric. To compute "CLIPSIM-*scenes*", abbr. "CLIPSIM-*s*", we calculate the clip image similarity for each frame and the given scene images. To compute "CLIPSIM-*frames*", abbr. "CLIPSIM-*f*", we calculate the clip score in between generated frames. Likewise, we utilize the clip text-image similarity as "CLIPSIM-*text*", abbr. "CLIPSIM-*t*", to quantify the semantic correlation between the generated videos and their corresponding descriptive text. In our experiments, we employ the MSR-VTT dataset due to its open-domain nature with textual annotations. We randomly selected one caption per video from the official test set (a total of 2,990 videos). In order to evaluate the quality of the generative transition, we considered the given scene images as two frames of a video with a larger interval timestep. This allows us to analyze and compare the smoothness and coherence of the transition between the two images. Specifically, we utilize the first and 64th frames as the reference scene images in our evaluation. Tab. 1 presents the comparative results of our analysis. The results demonstrate that our method outperforms the comparison algorithms. It is worth noting that our method does not achieve significantly higher scores than CLIPSIM-Scene and CLIPSIM-Frames. To further scrutinize the quality of our video transitions, we calculate FVD and frame-wise FID score. The outcomes reveal that our model consistently delivers superior video quality compared to other methods examined in our study.

Table 1: Comparison with methods *w.r.t.* CLIPSIM-*t*, CLIPSIM-*s*, CLIPSIM-*f*, FVD and FID.

| Methods | CLIPSIM-$t$ (↑) | CLIPSIM-$s$ (↑) | CLIPSIM-$f$ (↑) | FVD (↓) | FID (↓) |
|---|---|---|---|---|---|
| Morphing | 0.2535 | 0.7707 | 0.9569 | 573.9 | 35.3 |
| VQGAN-based Transition | 0.2528 | 0.7389 | 0.9542 | 445.6 | 36.7 |
| SD-based Transition | 0.2665 | 0.6080 | 0.8809 | 502.0 | 62.0 |
| Ours | **0.2726** | **0.7740** | **0.9675** | **181.6** | **13.4** |

## 4.3 DIVERSE AND CONTROLLABLE TRANSITION GENERATION

We showcase the advantages of our method in handling generative transitions: **1)** diverse results by sampling multiple times; and **2)** controllable results of our method by employing textural descriptions such as camera movement control. **Diverse Transition Generation.** We present visual results demonstrating the variety of transitions generated under identical prompts within two equivalent scenes, as depicted in Fig. 6. We can observe that as the raccoon transitions from being in the sea to the stage scene, the first two outcomes depict a form of similarity transition. However, the sequence of transitioning between the objects and backgrounds varies. The final transition, indeed, resembles the unfurling of a curtain, akin to a wave sweeping in from the right side. **Controllable Transition Generation.** Our method can achieve transitions by controlling the prompt with camera movements. For instance, by employing camera zoom-in and zoom-out control, we create visual occlusions, and during the zoom-out phase, we generate the subsequent scene to achieve a seamless scene transition. This occlusion technique is a widely employed transition method in filmmaking. Fig. 7 illustrates our example of this technique through the prompt, transitioning a scene from an office to a library with the presence of a panda.

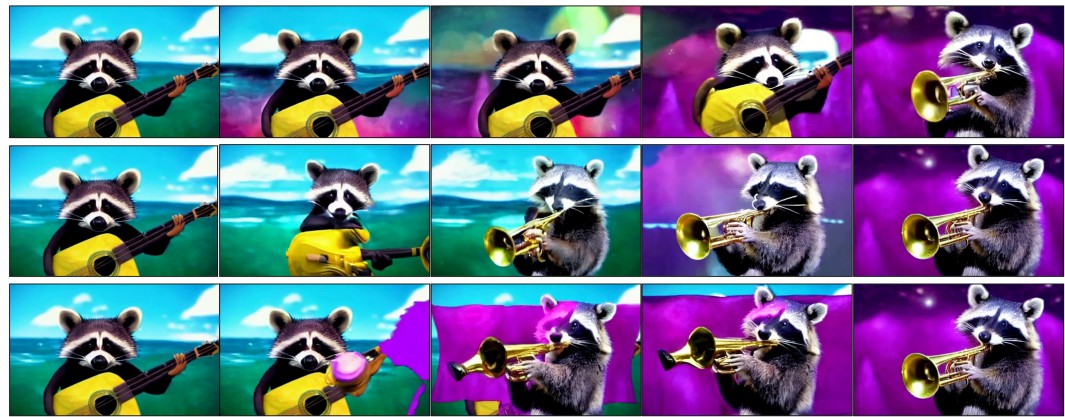

Figure 6: **Diverse transition results**. Each row illustrates frames of our transition results by *"the scene transition from the sea into a playing trumpet raccoon"*.

*1ˢᵗ prompt: A panda working in the office, camera zoom-in.*

*2ˢᵗ prompt: A panda working in the library, camera zoom-out.*

Figure 7: **Text-controllable transition.** Examples of our scene transition are achieved by controlling them through textual descriptions, i.e., camera-motion control.

## 4.4 COMPARISON WITH STATE-OF-THE-ART METHOD FOR VIDEO PREDICTION

We conducted a comprehensive comparative analysis between our proposed method and the state-of-the-art method TATS for the task of generating long videos consisting of 1024 frames, on the UCF-101 dataset (Soomro et al., 2012). Qualitative and quantitative comparisons are illustrated in Fig. 8. Fig. 8 (a) visually presents the comparative results, showing that TATS tends to degrade over time, leading to the emergence of grid-like artifacts and visual collapse. In contrast, our approach avoids such collapsing phenomena, maintaining stable and visually coherent video generation throughout the sequence. Fig. 8 (b) depicts the quantitative evaluation by FVD. In our study, we compared TATS with the outcomes obtained from our model's zero-shot generation and the results achieved after fine-tuning our model on the UCF-101 dataset. All models were evaluated under the auto-regressive setting. Our results are generated by the DDIM sampling of 100 steps. Our approach demonstrates a slower degradation of video quality over time compared to TATS. In the first few frames, due to the disparity of data distribution between the training set and UCF-101, our model's scores are higher than TATS. However, as TATS degrades over time, our model's FVD score becomes lower than that of TATS. Moreover, by performing fine-tuning on the UCF-101 dataset, our method consistently outperforms TATS.

## 4.5 MORE APPLICATIONS

**Long Video Generation.** Fig. 9 demonstrates the generation of long videos through auto-regressive prediction. Each video consists of tens of frames by auto-regressive prediction for 4 times. Notably,

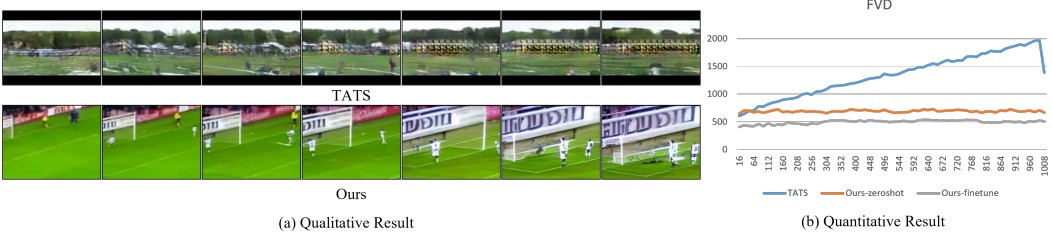

Figure 8: **Comparison for auto-regressive long video prediction.** a) Qualitative Results. Each frame is selected with a frame step 128 from the 256th frame; b) Qualitative results. The lower value indicates better quality.

*A raccoon dressed in a suit playing the trumpet, stage background.*     *A teddy bear washing the dishes.*     *Two pandas discussing an academic paper.*

Figure 9: **Long video results via auto-regressive prediction.** *Best view with Acrobat Reader. Click the images to play the videos.*

Ours       Gen-2       Ours       Gen-2

Figure 10: **Image animation results.** In each case, our result is on the left and the result of Gen-2 is on the right. *Best view with Acrobat Reader. Click the images to play the videos.*

our method ensures that the visual quality of the generated content remains pristine while preserving consistent semantic coherence throughout the synthesized results. **Image-to-Video Animation.** We demonstrate the outcomes of video generation by utilizing an image as the initial frame. This approach can be seamlessly integrated with other T2I models, such as Midjourney*. Here, we compare our results with those obtained from the online API image-to-video generation model, Gen-2 (Esser et al., 2023). From the comparison, it can be observed that despite not specifically training our model for this task, our model still achieves comparable results.

## 5 CONCLUSION

In this paper, we introduce **SEINE**, a S2L video diffusion model that focuses on smooth and creative transitions and auto-regressive predictions. SEINE utilizes a random-mask video diffusion model to automatically generate transitions based on textual descriptions, ensuring coherence and visual quality. The model can also be extended to image-to-video animation and auto-regressive video prediction tasks, allowing for the creation of coherent "story-level" long videos. The evaluation of transitions is conducted based on temporal consistency, semantic similarity, and video-text semantic alignment. The experimental results demonstrate the superiority of our approach over existing methods in terms of transition and prediction.

---

*https://www.midjourney.com

## 6 ETHICS STATEMENT

We acknowledge the ethical concerns that are shared with other text-to-image and text-to-video diffusion models (Ramesh et al., 2021; 2022; Rombach et al., 2022; Saharia et al., 2022; Ho et al., 2022; He et al., 2022; Wang et al., 2023b). Furthermore, our optimization is based on diffusion model (Ho et al., 2020; Song et al., 2021b), which has the potential to introduce unintended bias as a result of the training data. On the other hand, our approach presents a short-to-long video diffusion model that aims to create a coherent story-level long video, which may have implications for various domains including filmmaking, video games, and artistic creation.

## ACKNOWLEDGEMENTS

This work is supported by the National Key R&D Program of China under Grand NO.2022ZD0160100, the National Natural Science Foundation of China under Grant No. 62102150, the Ministry of Education Singapore under its MOE AcRF Tier 2 (MOE-T2EP20221-0012) and NTU NAP, and the Science and Technology Commission of Shanghai Municipality under Grant No. 23QD1400800, No. 22511105800, and the RIE2020 Industry Alignment Fund – Industry Collaboration Projects (IAF-ICP) Funding Initiative, as well as cash and in-kind contribution from the industry partner(s).

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

## A  APPENDIX

### A.1  COMPARED METHODS FOR TRANSITION.

Here we describe the details and implementation details for the compared methods achieving transition.

- **Morphing**. Morphing is a special effect in motion pictures and animations that changes (or morphs) one image or shape into another through a seamless transition. Traditionally such a depiction would be achieved through dissolving techniques on film. The transition is achieved by using the first and the last frames and generating the intermediate frames.

- **SD-based Method.** We adopt the SD as our baseline model. Since SD cannot directly generate intermediate frames, we first compute the latent code of the given two images by reverse diffusion process (*i.e.*, DDIM inversion (Song et al., 2021a)), then we use the interpolation of Euclidean distance of the latent codes to denoising and obtain the intermediate frames.

- **VQGAN-based Method.** VQGAN serves as the latent code layer for SD. We employ the interpolated representation of the codes from the encoder of VQGAN output (prior to quantization) as a reference method.

### A.2  FAILURE CASES

**Watermarks.** Since part of our training data (WebVid10M) contains watermarks, there is a chance that generated videos may also exhibit such signals (see Fig. 11 and 12). In the prediction task, we observed that in the autoregressive process, watermarks gradually emerge. Once they appear, they persist throughout the entire video. In the transition task, if the watermarks are present in the given images, the generated intermediate frames will also be affected. However, if no watermarks appear in the given images, watermarks are less likely to appear, and even if they do, they are usually faint.

*There is a table by a window with sunlight streaming through illuminating a pile of books.*     *A view of lake in autumn.*     *A panda playing on a swing set.*

Figure 11: **Failure cases for prediction with watermarks.** *Best view with Acrobat Reader. Click the images to play the videos.*

**Abrupt Transition.** While our method excels in transition generation, it does have limitations regarding the requirement for the similarity of the source and target scenes. A natural and smooth transition relies on finding meaningful correspondences between the two scenes. If the scenes are too dissimilar or lack common elements, the transition may appear abrupt or visually inconsistent. An alternative solution to address this limitation is through a multi-step approach. We utilize image editing methods (Tumanyan et al., 2023; Cao et al., 2023; Zhang & Agrawala, 2023) to transform the source image in the scenes and actions into the target image. This method is particularly beneficial in scene generation as it operates at the image level, naturally preserving similarities such as the structure, background, and objects of the image. As a result, the transition effects produced using this approach often appear highly natural.

**Text-video Unalignment.** In our S2L model, we aim to optimize $p_\theta(z_0|c, \tilde{z}_0)$ which $\tilde{z}_0$ represents the latent code of the unmasked condition image. However, in our training dataset, videos $z_0$, unmasked frames $\tilde{z}_0$, and captions/prompts ($c$) are not completely independent and exhibit strong correlations. Consequently, when there is no inherent connection between the given unmasked image and caption, it can lead to discontinuities or difficulties in aligning with video-text alignment.

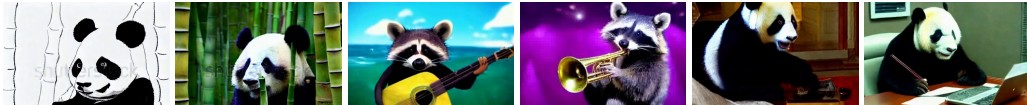

*A panda is in a bamboo forest, shifting from sketch-like to realistic style.* *The scene has changed from the sea into a playing trumpet raccoon.* *The scene has changed from the panda watching television to the panda working in the office.*

Figure 12: **Failure cases for transition with watermarks.** *Best view with Acrobat Reader. Click the images to play the videos.*

Additionally, we have observed that the quality of generated videos would be influenced to some extent by the given prompt. Providing prompts that align well with the realistic scenario and have a resemblance to unmasked videos can yield better results. Ensuring the provision of more suitable prompts in the context of generating long videos is a topic we will explore in our subsequent research.

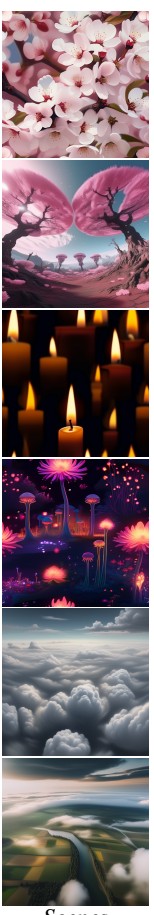

*Scenes*      *Transition Video 1*      *Transition Video 2*      *Transition Video 3*

Figure 13: **Transition samples.** In each case, transitions are generated by the same prompt with different random seeds. *Best view with Acrobat Reader. Click the images to play videos.*

### A.2.1 IMAGE-TO-VIDEO ANIMATION

We present additional outcomes of image-to-video animation, as depicted in Fig. 14. For each instance, the reference image is displayed on the left, while the corresponding animation video is showcased on the right. Notably, our model demonstrates the ability to accommodate images of various sizes. The figure showcases the animation results obtained from both square and rectangular images. The reference images in the first three rows are generated from Midjourney. The reference images in the last two rows are generated from Stable Diffusion XL.

*(a) First Frame is generated from Midjourney*

*(b) First Frame is generated from Stable Diffusion XL*

Figure 14: **Image-to-video animation results.** *Best view with Acrobat Reader. Click the images to play the transition videos.*

