# OpenReview forum: "SEINE: Short-to-Long Video Diffusion Model for Generative Transition and Prediction"
_ICLR.cc/2024/Conference — ICLR 2024 poster_

### Official Review · Reviewer_4xUi · 2023-10-12

**Soundness:** 2 fair
**Presentation:** 3 good
**Contribution:** 2 fair
**Rating:** 6
**Confidence:** 4

**Summary:**

This paper proposes a method to train a diffusion model with random masking on the frame level to enable video generation, prediction, and interpolation. They demonstrate that their method is able to generate longer video and create smooth transitions between two different frames. The authors demonstrate that their method outperforms baseline methods.

**Strengths:**

- The paper is well written, and clear
- The paper shows good long video generations, and is able to generate complex transitions between semantically different frames

**Weaknesses:**

My primary concerns center around the lack of baselines and novelty. Particularly, the authors fail to cite a few very related works, that accomplish similar tasks that enable frame prediction and interpolation.

- MaskViT [1], MAGVIT [2]: MaskGit-like models trained on tokenized video frames. Given the masked learning object, these models can also usually generalize to enable generation, prediction, and interpolation. MAGVIT is trained explicitly to do this.
- MCVD [3], RaMViD [4]: These two methods seem nearly identical to the proposed method, where a video diffusion model is trained with masked latents. An exception is a lack of text-conditioning and scale in [3,4], however, I do not believe that meets the bar as a point of novelty.

Could the authors please clarify on how their method is novel over the prior work mentioned above? In addition, it would be necessary to compare against a subset of these methods as baselines (or a similar model), as currently there are no baselines explicitly trained for the prediction / interpolation tasks.

[1] Gupta, Agrim, et al. "Maskvit: Masked visual pre-training for video prediction." arXiv preprint arXiv:2206.11894 (2022).

[2] Yu, Lijun, et al. "Magvit: Masked generative video transformer." Proceedings of the IEEE/CVF Conference on Computer Vision and Pattern Recognition. 2023.

[3] Voleti, Vikram, Alexia Jolicoeur-Martineau, and Chris Pal. "MCVD-masked conditional video diffusion for prediction, generation, and interpolation." Advances in Neural Information Processing Systems 35 (2022): 23371-23385.

[4] Höppe, Tobias, et al. "Diffusion models for video prediction and infilling." arXiv preprint arXiv:2206.07696 (2022).

**Questions:**

See weaknesses

---

> ### Author Response · Authors · 2023-11-23
> **Response from the Authors**
>
> **Q1 Lack of Novlety**
>
> **A1** We will include the listed works in the Related Works in the revised manuscript
>
> We delve into the study of a new task, termed as "generative transition", which aims to produce diverse and creative video segments to bridge two distinct scenes. To this end, we propose a short-to-long video diffusion model as an efficient solution, utilizing the initial and final frames of a given scene as inputs. Our model incorporates a text-based and video conditioning method to generate smooth transitional videos. This approach enables us to effectively manage the transition process, ensuring both the coherence and visual quality of the generated videos. Unlike traditional methods that rely on a pre-defined effect or simple interpolation for conditional transition effects, our model can handle more complex scenes and allows for controllable text prompts. We believe this method will inspire greater creativity in video production creators and the industry, while also presenting an intriguing research topic for the community.
>
> We summarize our contributions as follows:
> 1) We propose a new task, namely generative video transition and prediction, aiming at coherent “story-level” long video generation with smooth and creative scene transition and varying lengths of videos;
> 2) We adopt the random-mask-based diffusion model for generating smoothness transition video;
> 3) we propose three assessing criteria for transition and extensive experiments demonstrate our method the superior performance on the metrics, as well as its ability to be applied across versatile applications
>
> We argue that Masking is a general technique, and has been first used in NLP, e.g., BERT[1]. MCVD and RaMViD also adopted this technique and applied it to diffusion-based video generation. Deviating from them, we propose a latent-mask-based approach, which is more efficient than pixel-based approaches (e.g., MCVD and RaMViD). SEINE is able to conduct masking in more compressed space and directly learn the distribution of entire latent spatiotemporal distribution rather than explicit video distribution.
>
> [Reference]
>
> [1] Devlin J, Chang M W, Lee K, et al. Bert: Pre-training of deep bidirectional transformers for language understanding[J]. arXiv preprint arXiv:1810.04805, 2018.
>
> **Q2 In addition, it would be necessary to compare against a subset of these methods as baselines (or a similar model), as currently there are no baselines explicitly trained for the prediction/interpolation tasks.**
>
> **A2.**
> Thank you for the constructive suggestion. In our paper, we compared our model's predictions in Fig. 8 (b) with those of TATS, a state-of-the-art method for video prediction. Our methods outperformed the compared method in terms of overall quality and degradation. To perform further analysis with the random mask-based method, we used MCVD as a baseline for comparison. We adhered to the predictive settings on UCF-101. We sampled 16 frames from both our model and MCVD for comparison. The results shown below demonstrate that our method achieves a better FVD than MCVD.
> |  Method| MCVD | Ours |
> | --- | --- | --- |
> | FVD | 1143 | 406 |

---

> > ### Comment · Reviewer_4xUi · 2023-12-01
> > **Response**
> >
> > Thank you for the rebuttal. My general concerns about novelty / contribution have been mostly addressed, and I appreciate the comparison between MCVD / RamViD (pixel-based) and SEINE (latent space). However, I still think the contribution with respect to architecture is still relatively low, as masking in pixel vs latent space is a relatively small difference, and performance gains seem to be mainly from using a latent space, which is a commonly known (e.g. LDM).
> >
> > After considering the other reviews, as well as rebuttals, I will raise my score to a 6. In general, results look good, and generating transitions using a video model seems like a promising approach.

---

### Official Review · Reviewer_Yzaz · 2023-10-22

**Soundness:** 3 good
**Presentation:** 3 good
**Contribution:** 3 good
**Rating:** 6
**Confidence:** 4

**Summary:**

The paper proposes a new problem of generative transition and prediction, which can help generate story-level videos through different shot transitions. The author also proposed a short-to-long video diffusion model, which utilizes a random mask strategy for training. To evaluate the task, this paper proposes three assessing criteria. Both objective and subjective evaluation prove their proposed method’s effectiveness.

**Strengths:**

This article addresses the limitation of existing models that can only generate shot-level videos and proposes a method to generate story-level videos using transitions. They extend an existing video generation framework and achieve impressive results in generating long videos. They also propose a reasonable evaluation framework to assess the proposed model, and a large number of demos and quantitative evaluations demonstrate the effectiveness of their approach. The contribution of this work is significant.

**Weaknesses:**

The author should provide a more detailed description of the model for reproducibility, including training resources, training parameters, and so on. Additionally, the author should also report scores on commonly used evaluation metrics such as FID.

**Questions:**

I wondered how many GPUs they used and how long it takes for training. Besides, as far as I know, FID is used to evaluate video generation quality in many papers, can they provide this to make their paper more solid?

---

> ### Author Response · Authors · 2023-11-23
> **Response from the Authors**
>
> **Q1. The author should provide a more detailed description of the model for reproducibility, including training resources, training parameters, and so on.**
>
> **A1.** Thank you for your suggestion. Our model is trained on the video-text dataset Webvid10M dataset on 16GPUs A100 for two weeks. To eliminate the watermark, the model is trained on the internal watermark dataset for two days. The video clips input into the model are 16 frames, obtained by sampling at intervals of 6 frames. In all the training stages, we use a learning rate of 1×10−4 and AdamW optimization. We have prepared our model and code and will release them after the anonymous review phase. We hope our model and code could contribute to the open-source community for open-would video transition and prediction.
>
> **Q2. Additionally, the author should also report scores on commonly used evaluation metrics such as FID. Besides, as far as I know, FID is used to evaluate video generation quality in many papers, can they provide this to make their paper more solid?**
>
> **A2.** Thank you for the suggestion. We conducted FVD and FID to evaluate the video generation quality of our model. The results are shown in the table below. As we demonstrate, our method outperforms the compared method.
>
> | Method  | Morphing | VQGAN-based | SD-based | Ours |
> | --- | --- | --- | --- | --- |
> | FVD | 583.9 | 445.6 | 502.0 | 181.6 |
> | FID | 34.3 | 36.7 | 62.0 | 13.4 |
>
> **Q3. I wondered how many GPUs they used and how long it takes for training.**
>
> **A3.**  Our model is trained on the video-text dataset Webvid10M dataset on 16GPUs A100 for two weeks. To eliminate the watermark, the model is trained on the internal watermark dataset for two days.

---

> > ### Comment · Reviewer_Yzaz · 2023-12-01
> > **Thanks for your responses**
> >
> > Thanks for the effort and details for making these revisions. Additional evaluation results show that the proposed method performs well. My concerns have been satisfactorily addressed.

---

### Official Review · Reviewer_Tv5a · 2023-11-01

**Soundness:** 3 good
**Presentation:** 3 good
**Contribution:** 2 fair
**Rating:** 5
**Confidence:** 3

**Summary:**

In this paper, the author focus on a new task, "generative transition", which aims at smooth and creative transitions between scenes. Specifically, this paper proposes SEINE, a short-to-long video diffusion model with random masks to generate transitions frames based on textual prompts that describe transitions. Given a few unmasked frames, the proposed random-mask based diffusion model is able to generate frames at arbitrary positions. Therefore, their model can be used for tasks including generative transition, long video generation, and image-to-video animation by giving unmasked frames at different positions.

In the experiments, the authors compare their model with other baselines including morphing, VQGAN-based transition, and SD-based transition. Quantitative and qualitative results show that SEINE has better transition temporal coherence, semantic similarity across frames, and better video-text alignment.

**Strengths:**

- The task of generative transition is novel and rarely explored, which I believe is one of the main novelty of this paper. As current text-to-video generation models are mostly tacking short video clips, a smooth and creative transition between these short clips is of increasing importance.

- The proposed random-mask based model seems a good solution for this task. In addition to generating transition frames, the random-mask based model can also deal with long-video generation and image-to-video generation by giving unmasked frames at different positions.

- The quantitative and qualitative experiments demonstrate the effectiveness of the proposed model.

**Weaknesses:**

- From the qualitative result shown in Figure 6, it seems that the transition is more like a "interpolation" between two scenes. For example, the frames in (row1, col4). (row2, col2), and (row2, col3) are not very natural.

- In Figure 5 right part (the cat example), it seems that morphing also provides a descent transition. So for two frames with small transitions needed in between, it seems that the proposed method might add unnecessary variety/creativity.

- In general, it's hard to see if the proposed method provides a good solution to this new task. The paper also lacks enough ablation study of the model architecture design. More discussions and intuitions about this task would be helpful for future works.

- Some minor things:
1. In Sec. 2, the citation for PYoCo is missing.
2. In the last paragraph of Sec. 3, the sentence describing "Long video" is incomplete.
3. Figure 4 is not easy to understand at first glance. It would be nice to add more descriptions for better readability.
4. In Figure 10, the image on the left part has red-green-blue watermarks. Is that example from Gen-2 instead of SEINE?

**Questions:**

- For controllable transition generation, do we give the first and last frames unmasked to the modl for each prompt? If this is the case, I'm wondering maybe the model can also generate smooth zoom-in/out transitions without explicitly adding "camera zoom-in/out" in the prompt. It would be nice to provide ablation study that removes "camera zoom-in/out" in the prompts and see if the generation quality deteriorates.

- As mentioned in the above part, it would be nice if the author can provide some discussions about what kinds of scene transitions (small transition vs large transition, same object vs different objects) their model is good at.

**Details Of Ethics Concerns:**

No ethics concerns.

---

> ### Author Response · Authors · 2023-11-23
> **Response from the Authors （Q1&Q2)**
>
> **Q1. From the qualitative result shown in Figure 6, it seems that the transition is more like a "interpolation" between two scenes. For example, the frames in (row1, col4). (row2, col2), and (row2, col3) are not very natural.**
>
> **A1. Re More like a "interpolation" :**
> For Figure 6, our results differ significantly from those obtained through interpolation. In the newly uploaded rebuttal.pdf of Fig. R3, we have included animated gif results for better visualization. These results demonstrate our approach's diversity and creativity. For example, in sample 1, a raccoon appears to sway to the right along with the ocean waves before morphing into a figure wearing a trumpet. Sample 2 showcases a scene transition achieved through the shimmering effects of stage lighting. Meanwhile, sample 3 depicts a transition from the ocean to the stage, as if unveiled by a theater curtain. In contrast to the interpolation methods (refer to the first row in Fig. R4), the VQGAN-based and Morphing techniques seem to merely employ a simplistic fade-in, fade-out approach.
>
> **Re unnatural results:**
> We recognize the presence of motion distortion in the intermediate frames during video transitions. Viewing these frames in isolation might seem unnatural, as they are transitional states. However, we've observed that within the video format in Fig. R3, these distortions integrate smoothly, resulting in natural transition frames when viewed as part of the entire video sequence.
>
> **Q2.In Figure 5 right part (the cat example), it seems that morphing also provides a descent transition. So for two frames with small transitions needed in between, it seems that the proposed method might add unnecessary variety/creativity.**
>
> **A2.**   We acknowledge that for minor transitions, our method may not differ significantly from morphing techniques. Small transitions may be handled using interpolation or morphing. However, for larger transitions, like scene changes, interpolation often results in blurred effects and fails to create meaningful or creative content. In these scenarios, a generative video model is necessary for producing effective transition models. We have performed additional experiments on large transitions, as depicted in Fig. R1, and compared them with results from Fig. R2. These results illustrate that our transitions are not only natural but also capable of generating diverse outcomes.

---

> ### Author Response · Authors · 2023-11-23
> **Response from the Authors (Q3)**
>
> **Q3. In general, it's hard to see if the proposed method provides a good solution to this new task. The paper also lacks enough ablation study of the model architecture design. More discussions and intuitions about this task would be helpful for future works.**
>
> **A3. Re Ablation study of the model architecture design:**
> We add an ablation study of our model design. Our primary focus is on the exploration of a rarely studied task: generative video transition. To address this, we introduce a random mask layer into the base T2V model, following the architecture proposed by Lavie (Wang et al., 2023). This architecture incorporates a stable diffusion v1.4 and temporal attention layers.
> To assess the effectiveness of the introduced random mask layer, we conducted an ablation study on the zero-shot UCF-101 dataset. In this study, we utilized fixed masks to mask the first one or two frames. We autoregressively sampled the first 60 frames from a set of 10,100 video samples in the UCF-101 dataset and used Fréchet Video Distance (FVD) as the evaluation metric for the models.
> Our findings reveal that our model outperforms fixed mask models, indicating that the random mask model exhibits greater robustness. Additionally, our random mask model demonstrates the ability to perform Image-to-Video generation, generate long videos, and execute video transitions within a single model.
>
> |fvd	|mask 1 frame	|mask 2 frame2|
> |---|---|---|
> |fixed mask|	683.4	|733.3|
> |random mask(ours)	|672.7	|727.3|
>
> **Re More Discussion:**
> In addition to the failure cases we provided in the paper,  we will add discussions about the following two points in our final version:
>
> 1) Large transition and small transition: our model is capable of small transitions of similar scenes for a single object and also the large transition of different scenes. While small transitions may also be achieved by interpolation or morphing methods, our model can also generate creative and diverse results. For a more comprehensive explanation and discussion, please refer to the responses provided for Question 1 and Question 6.
>
> 2) text-controllable transition: The generated transition effect is influenced by both the input image and the prompt. When there is an inherent connection between the input images, the model automatically identifies similarities and generates transitions accordingly, even when the prompt is reduced or omitted. However, when specific prompt requirements are introduced, the model generates transitions based on those requirements. We conducted an ablation study to investigate controllable transition generation. For a more comprehensive explanation and discussion, please refer to the response provided for Question 5.
>
> **Re Intuitions:**
> We delve into the study of a new task, termed as "generative transition", which aims to produce diverse and creative video segments to bridge two distinct scenes. To this end, we propose a short-to-long video diffusion model as an efficient solution, utilizing the initial and final frames of a given scene as inputs. Our model incorporates a text-based and video conditioning method to generate smooth transitional videos. This approach enables us to effectively manage the transition process, ensuring both the coherence and visual quality of the generated videos. Unlike traditional methods that rely on a pre-defined effect or simple interpolation for conditional transition effects, our model can handle more complex scenes and allows for controllable text prompts. We believe this method will inspire greater creativity in video production creators and the industry, while also presenting an intriguing research topic for the community.

---

> ### Author Response · Authors · 2023-11-23
> **Response from the Authors (Q4&Q5)**
>
> **Q4. Some minor things:**
> 1. In Sec. 2, the citation for PYoCo is missing.
> 2. In the last paragraph of Sec. 3, the sentence describing "Long video" is incomplete.
> 3. Figure 4 is not easy to understand at first glance. It would be nice to add more descriptions for better readability.
> 4. In Figure 10, the image on the left part has red-green-blue watermarks. Is that example from Gen-2 instead of SEINE?
>
> **A4.1** Thank you for the reminder. PYoco (Ge et al., 2023) is an advanced text-to-video generation model that presents a noise prior approach and utilizes a pre-trained eDiff-I (Balaji et al., 2022) as initialization. We will add the reference in the updated version.
>
> [Reference]
>
> Songwei Ge, Seungjun Nah, Guilin Liu, Tyler Poon, Andrew Tao, Bryan Catanzaro, David Jacobs, Jia-Bin Huang, Ming-Yu Liu, and Yogesh Balaji. Preserve your own correlation: A noise prior for video diffusion models. ICCV 2024.
>
> Yogesh Balaji, Seungjun Nah, Xun Huang, Arash Vahdat, Jiaming Song, Karsten Kreis, Miika Aittala, Timo Aila, Samuli Laine, Bryan Catanzaro, et al. ediffi: Text-to-image diffusion models with an ensemble of expert denoisers. arXiv preprint arXiv:2211.01324, 2022
>
> **A4.2** Thank you for your reminder. We apologize for the typo and we will complete the sentence in our updated version: "Long video involves recursive using the last few frames of the generated video as input and utilizing masks to predict the subsequent frames. By recursively iterating this process, the model can generate longer video sequences.
>
> **A4.3** Thank you for the suggestion.  We will add more descriptions to the captions of Figures for better readability: "During training, the clean video $x_0$ is encoded into a latent code $z_0$ by a pre-trained encoder. The model is then fed with a concatenation of a randomly masked latent code, corrupted code $z_t$, and masks. During inference, transition results can be obtained by inputting the concatenation of noise, masked first and last frames of the video, and the masking tensor $\tilde{m}_0$. Prediction results can be achieved by inputting the concatenation of noise, the first masked frame tensor, and the mask tensor."
>
> **A4.4** In Figure 10, we show a comparison of our results and Gen-2. In each case, our result is shown on the left and the results of Gen-2 are shown on the right. We found that our model reaches comparable results for image-to-video animation.
>
> **Q5. For controllable transition generation, do we give the first and last frames unmasked to the model for each prompt? If this is the case, I'm wondering maybe the model can also generate smooth zoom-in/out transitions without explicitly adding "camera zoom-in/out" in the prompt. It would be nice to provide ablation study that removes "camera zoom-in/out" in the prompts and see if the generation quality deteriorates.**
>
> **A.5** Thank you for your question and suggestion. We have added an ablation study for controllable transition generation. We find that the effect of the transition is determined jointly by the given image and the prompt. In the example of the panda, even when we remove the zoom in/zoom out prompt, we still observe that the transition video is performing a zoom-in. This is because the ratio of the panda in the video changes between the two given frames. To achieve a smooth transition, the model tends to zoom in during generation. Additionally, for further research, we used more complex images without a clear single subject for transitions, and the results are shown in Fig. R5. We can see that the transition effects can be somewhat controlled based on different prompts.

---

> ### Author Response · Authors · 2023-11-23
> **Response from the Authors （Q6)**
>
> **Q6. As mentioned in the above part, it would be nice if the author can provide some discussions about what kinds of scene transitions (small transition vs large transition, same object vs different objects) their model is good at.**
>
> **A6.** Thank you for your constructive comment.  Compared to the previous method, we summarize our model is good at generating complicated scenes and large transitions.
> For large transitions involving complex scenes, our model performs exceptionally well when compared to previous methods. It can generate diverse and creative video segments, effectively bridging distinct scenes. This proficiency is evident from the results demonstrated in Fig. R1, where it managed a variety of objects with ease and creativity.  We compared our model against other methods, which are illustrated in Fig. R2. This comparison further highlights the strengths of our approach over conventional techniques.
>
> When it comes to handling transitions involving the same object but within different scenes, as depicted in Fig. R3,  We found our model is good at handling different scenes. In contrast, Morphing and VQGAN interpolation-based method output videos often result in blurred effects and fail to create meaningful or creative content, and the SD-based method struggles to produce videos with temporal coherence.

---

### Official Review · Reviewer_GChA · 2023-11-01

**Soundness:** 3 good
**Presentation:** 3 good
**Contribution:** 3 good
**Rating:** 5
**Confidence:** 3

**Summary:**

The paper presents the SEINE model, for generating "story-level" long videos from short clips. It introduces a unique problem in generative transition and prediction. Using a random-mask video diffusion based on textual descriptions, the model shows smooth transitions between scenes. To evaluate its efficacy, the authors provide three new criteria: temporal consistency, semantic similarity, and video-text alignment. Results show its potential for generating coherent long videos.

**Strengths:**

- The method of using masks was proposed in [1] and [2], but as far as I know, this is the first time it has been used in video transition. It could be novel.

- The proposed method shows better performance on the metric compared to the baseline.

- The proposed method can be applied in various areas such as long video generation and image-to-video animation.


References

[1] Voleti, Vikram, Alexia Jolicoeur-Martineau, and Chris Pal. "MCVD-masked conditional video diffusion for prediction, generation, and interpolation." Advances in Neural Information Processing Systems 35 (2022): 23371-23385.

[2] Fu, Tsu-Jui, et al. "Tell me what happened: Unifying text-guided video completion via multimodal masked video generation." Proceedings of the IEEE/CVF Conference on Computer Vision and Pattern Recognition. 2023.

**Weaknesses:**

- [Major] My main concern is that there is not enough quantitative evaluation of video transitions. This paper conducted quantitative experiments by randomly selecting one caption from MSRVTT and determining CLIP-TEXT. However, since video transitions occur when scenes change, it does not seem appropriate to evaluate video semantic correlation. Also, no video quality evaluation metrics (such as FVD etc.) have been considered. This makes it difficult to quantify the exact quality of generation.

- [Major] Several details related to the human evaluation are missing. (such as number of frames in the generated video, the dataset used, and the questions posed in the user study.)
Was the user study appropriately reflective of temporal coherence, text-video alignment, and semantic similarity?

- [Minor] For transitions to be applied in the real-world, it would require generating more than 16 frames. Would the quality be maintained if more frames are generated?

- [Minor] In Figure 5 related to video transition, the frame numbers and details are omitted.

**Questions:**

How long does the inference take? Is it capable of handling transitions with multiple objects across more than two scenes?

---

> ### Author Response · Authors · 2023-11-23
> **Response from the Authors (Q1~Q4)**
>
> **Q1. My main concern is that there is not enough quantitative evaluation of video transitions. This paper conducted quantitative experiments by randomly selecting one caption from MSRVTT and determining CLIP-TEXT. However, since video transitions occur when scenes change, it does not seem appropriate to evaluate video semantic correlation. Also, no video quality evaluation metrics (such as FVD etc.) have been considered. This makes it difficult to quantify the exact quality of generation.**
>
> **A1.** Video transition is a relatively unexplored area in research. In our approach, we tackle this task by providing the initial and final frames of the preceding and succeeding scenes, supplemented with text control. To evaluate the effectiveness of our approach, we utilize text-video alignment. While our model generates two distinct scenes, our primary goal is to facilitate a smooth transition where the generated video frames maintain semantic coherence between the two scenes.
> Simultaneously, devising a comprehensive measure of video quality remains a challenge. Therefore, we've incorporated human evaluation as an auxiliary means to assess overall video quality. Encouragingly, both the metrics, CLIPSIM-Scenes and CLIPSIM-Frames, show positive correlations with the results from human evaluations, indicating consistency between quantitative metrics and human perception.
>
> To further scrutinize the quality of our video transitions, we carried out FVD analysis along with frame-wise FID score calculations. The outcomes are presented in the table below. The table reveals that our model consistently delivers superior video quality compared to other methods examined in our study.
>
> Method| Morphing | VQGAN-based | SD-based | Ours |
> | --- | --- | --- | --- | --- |
> FVD | 573.9 | 445.6 | 502.0 | 181.6
> FID | 35.3 | 36.7 | 62.0 | 13.4
>
> **Q2. - [Major] Several details related to the human evaluation are missing. (such as number of frames in the generated video, the dataset used, and the questions posed in the user study.) Was the user study appropriately reflective of temporal coherence, text-video alignment, and semantic similarity?**
>
> **A2.** In our human evaluation, our primary objective is to ascertain the quality of the videos generated by our model. To maintain impartiality in this assessment, we selected a random assortment of results from our quantitative evaluation. These results were generated through rigorous experiments conducted on the MSR-VTT dataset and each video comprises 16 frames.
>
> For the user study, we presented the participants with pairs of videos - one generated by our model and the other by a comparative method. Rather than guiding their assessment, we encouraged the participants to evaluate the overall quality of each video independently. Subsequently, they were asked to cast their vote for the video they perceived as superior in quality within each pair.
>
> **Q3. For transitions to be applied in the real-world, it would require generating more than 16 frames. Would the quality be maintained if more frames are generated?**
>
> **A3.** Our model can potentially maintain quality while generating additional frames. We utilize a combination of prediction and transition techniques, guided by prompts, to ensure the desired transition effects. This approach has been successfully demonstrated in longer transitions, as seen in our long video demo on the website in Supplemental Material “Adventure of a panda” from 0:11 to 0:14. However, it's important to note that this method involves multiple sampling stages, which poses a challenge for efficiently generating high-quality transitions over an extended number of frames. The effectiveness of maintaining quality over more frames remains an interesting problem.
>
> **Q4. In Figure 5 related to video transition, the frame numbers and details are omitted.**
>
> **A4.** We appreciate your suggestion and will include this detail for clarity in the next version. We aimed for a fairly balanced selection of intermediate frames and chose the 5th, 9th, and 13th frames to showcase.

---

> ### Author Response · Authors · 2023-11-23
> **Responses from Authors (Q5)**
>
> **Q5. How long does the inference take? Is it capable of handling transitions with multiple objects across more than two scenes?**
>
> **A5.**
> **Re Inference Time:**
> The inference time depends on the used GPU and denoising timestep. In our paper, results are generated by 250 timesteps via DDPM inference. It takes around 50 seconds to sample a transition video in size of 312 x 520 by using Nvidia A100.
>
> **Re Transitions with Multiple objects:** In addition to the simple transition for a single object, we added more results for complicated transitions for multiple objects, and the results are shown in Fig. R1. As we can see, our model is able to generate reasonable, creative and diverse transitions across various scenes.
>
> **Re Transition across more than two scenes:** Our model is designed to create transition videos between two distinct scenes. For scenarios involving more than two scenes, this could potentially be managed by generating transitions for each pair of scenes sequentially.

---

### Author Response · Authors · 2023-11-23
**To all reviewers**

We deeply appreciate the diligent reviews and valuable feedback provided by all four reviewers. To further clarify our points and provide more insights, we have included additional visual results and comparisons in the  $\color{brown}{rebuttal.pdf}$ found in the Supplementary Material. We kindly request you to refer to this newly added PDF for a detailed view of our visual results. **(Best view with Acrobat Reader by clicking the images to play videos.)**

---

### Meta-Review · Area_Chair_upKV · 2023-12-06

**Metareview:**

(a) Scientific claims and findings

This paper seeks to create extended, story-level videos. It aims to achieve this by focusing on learning how to generate transitions between two video clips. Additionally, the authors have incorporated a random frame masking strategy into their video diffusion models, aiming to enhance the generation of transitions.

(b) Strengths

i. The task is novel and increasingly important in the realm of video generation.

ii. The paper presents compelling quantitative and qualitative results. The authors effectively showcase the model's ability to generate extended videos.

iii. The paper is well-written and effectively communicates its ideas.

(c) Weakness

i. The evaluation falls short in substantiating the enhancement of transition generation. The quantitative results and user study primarily gauge overall video quality rather than focusing on transitions. The qualitative examples also lack strong persuasion regarding improved performance. It's challenging to discern if the proposed method offers a superior solution to this new task.

ii. The technical novelty appears limited as the concept of video masking has been previously explored across several works.

**Justification For Why Not Higher Score:**

The evaluation provided doesn't adequately support the primary claim of the paper. Overall, it's challenging to ascertain how the approach truly achieves the goal of enhancing video transitions, both in terms of the approach itself and the evaluation conducted.

**Justification For Why Not Lower Score:**

The qualitative results demonstrate decent visual quality, and the authors offer a video demonstration showcasing the generation of long videos using the proposed model.

---

### Decision · Program_Chairs · 2024-01-16

Accept (poster)